# Highly Sensitive Plasmonic Structures Utilizing a Silicon Dioxide Overlayer

**DOI:** 10.3390/nano12183090

**Published:** 2022-09-06

**Authors:** Jakub Chylek, Petra Maniakova, Petr Hlubina, Jaroslav Sobota, Dusan Pudis

**Affiliations:** 1Department of Physics, Technical University Ostrava, 17. Listopadu 2172/15, 708 00 Ostrava-Poruba, Czech Republic; 2Department of Physics, Faculty of Electrical Engineering and Information Technology, University of Zilina, Univerzitna 1, 01026 Zilina, Slovakia; 3Institute of Scientific Instruments of the Czech Academy of Sciences, Kralovopolska 147, 612 64 Brno, Czech Republic

**Keywords:** surface plasmon resonance, Kretschmann configuration, silicon dioxide overlayer, reflectance, aqueous analyte sensing

## Abstract

In this paper, simple and highly sensitive plasmonic structures are analyzed theoretically and experimentally. A structure comprising a glass substrate with a gold layer, two adhesion layers of chromium, and a silicon dioxide overlayer is employed in liquid analyte sensing. The sensing properties of two structures with distinct protective layer thicknesses are derived based on a wavelength interrogation method. Spectral reflectance responses in the Kretschmann configuration with a coupling BK7 prism are presented, using the thicknesses of individual layers obtained by a method of spectral ellipsometry. In the measured spectral reflectance, a pronounced dip is resolved, which is strongly red-shifted as the refractive index (RI) of the analyte increases. Consequently, a sensitivity of 15,785 nm per RI unit (RIU) and a figure of merit (FOM) of 37.9 RIU${}^{-1}$ are reached for the silicon dioxide overlayer thickness of 147.5 nm. These results are in agreement with the theoretical ones, confirming that both the sensitivity and FOM can be enhanced using a thicker silicon dioxide overlayer. The designed structures prove to be advantageous as their durable design ensures the repeatability of measurement and extends their employment compared to regularly used structures for aqueous analyte sensing.

## 1. Introduction

The surface plasmon resonance (SPR) phenomenon has a wide range of applications in many branches of natural sciences for the purpose of sensing [1,2,3,4]. The SPR sensors utilize various techniques, such as intensity [5] or phase [6] detection based on either wavelength [7,8,9,10] or angular [8,11] interrogation methods. Amongst the most commonly used prism-based SPR excitation techniques is utilizing the attenuated total reflection (ATR) method in the Kretschmann configuration [1,6,7,8,9,12]. Due to high sensitivity to the refractive index (RI) of the surrounding environment, SPR is mainly exploited for RI sensing; however, the pivotal factors that determine the performance of the sensor are the structural and optical parameters of the employed SPR structure. Various structures that utilize a dielectric overlayer, such as a thin silicon dioxide film [13] or a tungsten disulfide nanosheet overlayer [14], have been designed to enhance the sensitivity or to increase the detection accuracy using a graphene–MoS${}_{2}$ hybrid structure [15]. The broad effect of coating the plasmonic material with a dielectric layer on sensing properties has been thoroughly studied for distinct materials and thicknesses, including a thin dielectric layer [16], a thin layer of silicon [17] or a semiconductor–metal–dielectric heterojunction system [18]. Moreover, the addition of a dielectric overlayer, such as silicon dioxide, in a simple plasmonic structure [13,19] or in complex structures [20,21], can lead to the excitation of guided modes that can greatly improve the performance of the sensor. In some instances, extremely sensitive plasmonic platforms based on metamaterials have been proposed and realized [22,23,24,25,26]. A porous nanorod layer has been utilized, where the achieved RI sensitivity attained a value of more than 30,000 nm/RIU [22]. Amongst other top performers, a miniaturized plasmonic biosensing platform based on a hyperbolic metamaterial using a grating-coupling technique has been reported [24], achieving RI sensitivity up to 30,000 nm/RIU and a figure of merit (FOM) of 590 RIU${}^{-1}$. Furthermore, a gold nanorod hyperbolic metamaterial-based sensor utilizing a prism coupling technique that could reach a sensitivity of 41,600 nm/RIU and FOM of 416 RIU${}^{-1}$ was reported in [26]. Moreover, using nanoporous gold films, RI sensitivity over 15,000 in the near-infrared range (NIR) has been reported in [25].

Generally, the design of the proposed structure proves to be advantageous as it is very easy to change the analyte. The protective layer ensures the repeatability of measurement and the durability of the structure. Since gold is one of the most commonly used materials to form the plasmonic layer in the SPR structures, it is inevitable for its surface to become quickly damaged when it is in direct contact with an aqueous analyte. It is more fragile to clean, and the optical response changes after a few measurements when aqueous analytes are frequently studied.

In this paper, we present highly sensitive plasmonic structures consisting of a BK7 glass substrate, a gold layer, two adhesion layers of chromium, and a protective layer of silicon dioxide. This is an improved structure in which a polymer layer [10] is substituted by a protective dielectric layer. Proposed structures are employed in the Kretschmann configuration with a coupling BK7 prism for aqueous analyte sensing. A method of spectral ellipsometry is utilized to determine the thicknesses of individual layers in the structure, including the silicon dioxide layer thicknesses of 147.5 nm and 270.9 nm. Using the obtained thickness parameters, the reflectance spectra for aqueous solutions of NaCl with distinct concentrations are calculated, and the sensing properties of the structures are derived. The first structure, with a thinner SiO${}_{2}$ protective layer, showed sensitivity to RI of the analyte up to 7130 nm/RIU, with FOM attaining a value of 25.7 RIU${}^{-1}$, and the second structure, with a thicker SiO${}_{2}$ protective layer, showed sensitivity to RI up to 7660 nm/RIU, with FOM attaining a value of 38.0 RIU${}^{-1}$. To confirm the theoretically derived values, measurements of the spectral reflectance ratio were performed for both structures. The following sensitivities to RI were achieved: for the first structure, up to 8140 nm/RIU, and for the second structure, up to 8690 nm/RIU. Furthermore, another measurement was performed for a smaller angle of incidence for the first structure with the silicon dioxide layer thickness of 147.5 nm, and RI sensitivity up to 15,785 nm/RIU was achieved, with FOM attaining a value of 37.9 RIU${}^{-1}$.

## 2. Theoretical Background

### 2.1. Structure Design

The structure under study consists of a BK7 glass substrate, adhesion layers of chromium, a gold layer, and a protective layer of silicon dioxide. A schematic drawing of the structure is shown in Figure 1.

### 2.2. Transfer Matrix Method

To express the optical response of the multilayer structures, the transfer matrix method (TMM) was utilized. We can describe the propagation of electromagnetic waves through a system of thin layers with the use of transmission and propagation matrices for each layer [27]:(1)$$D_{ij}=\frac{1}{t_{ij}}\left[\begin{array}{cc} 1 & r_{ij}\\ r_{ij} & 1 \end{array}\right],$$
where $D_{ij}$ is a transmission matrix at the *ij*-th interface (for i=j−1) and $P_{j}$ is a propagation matrix through the *j*-th layer:(2)$$P_{j}=\left[\begin{array}{cc} e^{ik_{j}t_{j}} & 0\\ 0 & e^{-ik_{j}t_{j}} \end{array}\right],$$
where $t_{j}$ is a layer thickness and $k_{j}$ is a wave-vector component perpendicular to the *ij*-th interface, and, for a prism of refractive index $n_{p}\left(\lambda \right)$ and an angle of incidence θ of the incident light, it can be expressed as:(3)$$k_{j}\left(\lambda \right)=\frac{2\pi }{\lambda }{\left[n_{j}^{2}\left(\lambda \right)-n_{p}^{2}\left(\lambda \right)\sin^{2}\theta \right]}^{1/2}.$$

The total transfer matrix M is then obtained by linking together the transmission and propagation matrices across the entire structure:(4)$$M=\left[\prod_{j=1}^{N}D_{ij}P_{j}\right]D_{N,N+1}=\left[\begin{array}{cc} M_{11} & M_{12}\\ M_{21} & M_{22} \end{array}\right],$$
where the reflection coefficient for *p*- and *s*-polarized light can be expressed as $r_{s,p}=\frac{M_{21}}{M_{11}}$.

### 2.3. Material Parameters

In this section, dispersion formulas used to model the optical response of the multilayer structures are listed with their corresponding parameters.

#### 2.3.1. Substrate (BK7 Glass) and Silicon Dioxide (${\mathrm{SiO}}_{2}$)

The refractive index of BK7 glass and ${\mathrm{SiO}}_{2}$ as a function of wavelength can be described by a three-term Sellmeier formula [28]:(5)$$n^{2}\left(\lambda \right)=1+\sum_{i=1}^{3}\frac{A_{i}\lambda^{2}}{\lambda^{2}-B_{i}},$$
where λ is the wavelength in μm and, for the BK7 glass, the values of the Sellmeier coefficient at room temperature are as follows: $A_{1}$ = 1.03961212, $A_{2}$ = 2.31792344 $\times \;10^{-1}$, $A_{3}$ = 1.01046945, $B_{1}$ = 6.00069867 $\times \;10^{-3}\;\mu m^{2}$, $B_{2}$ = 2.00179144 $\times \;10^{-2}\;\mu m^{2}$, and $B_{3}$ = 1.03560653 $\times \;10^{2}$$\;\mu m^{2}$. Similarly, for ${\mathrm{SiO}}_{2}$, the values of the Sellmeier coefficient at room temperature [28,29] are as follows: $A_{1}$ = 0.6961663, $A_{2}$ = 0.4079426, $A_{3}$ = 0.8974794, $B_{1}$ = 0.0684043 $\mu m^{2}$, $B_{2}$ = 0.1162414 $\mu m^{2}$, $B_{3}$ = 9.896161 $\mu m^{2}$.

#### 2.3.2. Gold Layer

The dispersion of the gold layer can be described by the complex dielectric function given by the Drude–Lorentz model with the parameters listed in Table 1 [30]:(6)$$\varepsilon_{\mathrm{Au}}\left(\lambda \right)=\varepsilon_{\infty }-\frac{1}{\lambda_{p}^{2}\left(1/\lambda^{2}+i/\gamma_{p}\lambda \right)}-\sum_{j=1}^{2}\frac{A_{j}}{\lambda_{j}^{2}\left(1/\lambda^{2}-1/\lambda_{j}^{2}\right)+i\lambda_{j}^{2}/\gamma_{j}\lambda }.$$

#### 2.3.3. Adhesion Chromium Layers

The dispersion of the adhesion chromium layers can be described by the complex dielectric function given by the Drude Critical Point (CP) model with the parameters listed in Table 2 [31,32]:(7)$$\varepsilon_{\mathrm{Cr}}\left(\lambda \right)=\varepsilon_{\infty }-\frac{1}{\lambda_{p}^{2}\left(1/\lambda^{2}+i/\gamma_{p}\lambda \right)}+\sum_{j=1}^{2}\frac{A_{j}}{\lambda_{j}}\left[\frac{e^{i\phi_{j}}}{\left(1/\lambda_{j}-1/\lambda -i/\gamma_{j}\right)}+\frac{e^{-i\phi_{j}}}{\left(1/\lambda_{j}+1/\lambda +i/\gamma_{j}\right)}\right]\;.$$

## 3. Theoretical Analysis

The optical properties and structural parameters of the proposed SPR structure are pivotal factors that determine the sensing properties of the structure. It is crucial to consider the thicknesses of individual layers and their effects on the properties of a dip in the reflectance spectrum that can be resolved by a spectrometer operating in a NIR spectral range of 1000–1900 nm. Firstly, the effect of the thickness of the gold layer $t_{\mathrm{Au}}$ is considered. The spectral reflectance of *p*-polarized light $R_{p}\left(\lambda \right)$ for distinct thicknesses of the gold layer is shown in Figure 2a,b for the structure with silicon dioxide thicknesses $t_{{\mathrm{SiO}}_{2}}=150$ nm and $t_{{\mathrm{SiO}}_{2}}=300$ nm, respectively, when the analyte is water. It can be observed that the resonance dip shifts towards longer wavelengths as the gold layer thickness decreases, and it is narrower for the silicon dioxide thickness of 300 nm. Moreover, the dip is too wide for the gold layer thickness in the range of 20–30 nm. By increasing the gold layer thickness, the dip becomes more narrow and more shallow. Therefore, the optimal gold layer thickness that serves as a middle ground between the depth and width of the dip was resolved as $t_{\mathrm{Au}}=38$ nm.

To investigate the influence of the silicon dioxide thickness $t_{{\mathrm{SiO}}_{2}}$ on the properties of the resolved dip, the reflectance spectra $R_{p}\left(\lambda \right)$ were calculated for the silicon dioxide thickness range of 100–300 nm. The calculated spectra are shown in Figure 3a, and it is apparent that the dip width decreases and the position of the dip shifts towards longer wavelengths with increasing thickness of the silicon dioxide. Furthermore, the thickness of the adhesion layer of chromium in the contact with the silicon dioxide also quite considerably affects the width and depth of the resolved dip. The spectral reflectances $R_{p}\left(\lambda \right)$ for distinct thicknesses of the chromium adhesion layer are shown in Figure 3b for the silicon dioxide thickness $t_{{\mathrm{SiO}}_{2}}=150$ nm. On increasing the thickness $t_{\mathrm{Cr}}$, surface plasmons become more damped, and the dip shows decreasing depth and increasing width. The optimal thickness of the chromium adhesion layer is as small as possible. However, the absence of the adhesion layer, as we confirmed experimentally, significantly impacts the longevity of the structure. Thus, we have chosen $t_{\mathrm{Cr}}=2$ nm.

To investigate the sensing properties of the structures, it is important to determine the RI sensitivity, which is defined as follows:(8)$$S_{n}=\frac{\delta \lambda_{r}}{\delta n},$$
where $\delta \lambda_{r}$ is a change in the resonant wavelength, which corresponds to the change in the position of the dip related to a change in the refractive index of the analyte δn. Furthermore, the performance of the SPR structures can be also evaluated in terms of the figure of merit, which is defined as the sensitivity $S_{n}$ divided by the full-width half-maximum (FWHM) of the dip. Taking into account the depth *D* of the dip, the definition can be further expanded as [33]:(9)$$\mathrm{FOM}=D\frac{S_{n}}{\mathrm{FWHM}}.$$

The theoretical RI sensitivity and FOM as a function of silicon dioxide layer thickness were calculated in order to estimate the performance of the structure. The calculated RI sensitivity is shown in Figure 4a and FOM in Figure 4b for the realistic thicknesses of the chromium adhesion layer.

Based on the theoretical results, it is evident that increasing the silicon dioxide overlayer thickness leads to increased RI sensitivity and FOM and that the increase in the chromium adhesion layer thickness leads to a minor improvement in RI sensitivity. However, it is accompanied by a decrease in FOM, which is more pronounced for a thicker silicon dioxide overlayer.

Furthermore, RI sensitivity and FOM were calculated as a function of the gold layer thickness. The dependences are shown in Figure 5a,b. It is evident that the considered angle of incidence greatly impacts both RI sensitivity and FOM. Moreover, it can be observed that RI sensitivity decreases with the increasing thickness of the gold layer, and that the FOM takes a maximum value, similarly as in [34], near the gold layer thickness of 39 nm.

## 4. Fabrication of Structures

Two variations of the structure with different thicknesses of the protective layer were manufactured. In the process, the substrates were coated homogeneously by a deposition technique based on radio frequency (RF) magnetron sputtering [35] with chromium, gold, and silicon dioxide. We have used a Cr target with 99.95% purity, and Au and SiO${}_{2}$ targets with 99.99% purity. All targets were 152 mm in diameter. To deposit the Cr and Au layers, the working gas of argon was used under the deposition pressure of 0.150 Pa with forward RF power of 150 W and 100 W, respectively, at 13.56 MHz. SiO${}_{2}$ coating was deposited in the mixture of Ar and O${}_{2}$ under the total deposition pressure of 0.250 Pa and partial oxygen pressure of 0.022 Pa with forward RF power of 720 W at 13.56 MHz.

To confirm the theoretical results, the method of spectral ellipsometry was utilized to determine the thicknesses of individual layers in the manufactured structures. Ellipsometry measurements were performed for the angles of incidence in a range of 40–70${}^{\circ }$ for both structures. The individual layer thicknesses obtained by the evaluation of the ellipsometric measurements are listed in Table 3.

## 5. Responses of Real Structures

Theoretical spectral reflectance responses in the Kretschmann configuration with a coupling BK7 prism were calculated for the angle of incidence $\theta =68^{\circ }$, when aqueous solutions of NaCl with concentrations in a range of 0–10 wt% with a step of 2 wt% were considered. The results are presented for both structures in Figure 6. Theoretical spectral reflectance ratios $R_{p}\left(\lambda \right)/R_{s}\left(\lambda \right)$ were calculated using the thicknesses of individual layers obtained by the method of spectral ellipsometry. It can be observed that the calculated reflectance spectra show well-pronounced dips, whose width is nearly constant and which exhibit a shift towards longer wavelengths with the increasing RI of the analyte.

The resonance wavelength as a function of the refractive index of the analyte and corresponding second-order polynomial fit for both structures is shown in Figure 7a. The shift in the resonant wavelength is approaching a value of 100 nm for the different refractive indices of the aqueous solutions of NaCl. To be more specific, for the refractive index values of the analyte in the range of 1.3330–1.3482, the resonant wavelengths in the range of 1220.5–1314.0 nm were calculated for the structure with the silicon dioxide layer thickness $t_{{\mathrm{SiO}}_{2}}=147.5$ nm (the first thickness), and the resonant wavelengths in the range of 1531.0–1632.8 nm were calculated for the structure with the silicon dioxide layer thickness $t_{{\mathrm{SiO}}_{2}}=270.9$ nm (the second thickness). The determined RI sensitivity at the angle of incidence $\theta =68^{\circ }$ for both structures is shown in Figure 7b. For the first silicon dioxide overlayer thickness, the derived RI sensitivity varies in the range of 5150–7130 nm/RIU, and for the second one in the range of 5720–7660 nm/RIU. The sensitivity can be also expressed in terms of the mass fraction of the solute, which, for the first structure, varies in the range of 7.9 to 10.8 nm/wt%, and for the second structure in the range of 8.7 to 11.6 nm/wt%. The structure with the thicker layer of silicon dioxide is more sensitive to the RI of the analyte.

The highest FOM calculated for the angle of incidence $\theta =68^{\circ }$ and the first silicon dioxide layer thickness attains a value of 25.7 RIU${}^{-1}$, and for the second silicon dioxide layer thickness attains a value of 38.0 RIU${}^{-1}$.

## 6. Experimental Analysis

The experimental setup used to measure the reflectance response of the structure and the refractive index (RI) sensing ability in the NIR spectral range is shown in Figure 8. The setup consists of a polychromatic light source (HL-2000, Ocean Optics), a collimating lens, a polarizer (LP-VIS050, Thorlabs), an analyzer (LPVIS050, Thorlabs), a spectrometer (FT-NIR ARCoptix), and a computer. An angular rotation desk with a goniometer [36], which is used to adjust the angle of incidence, is not shown in Figure 8. It can be observed that a white light source (WLS)—a halogen lamp—is used with a polarizer oriented $45^{\circ }$ with respect to the plane of incidence, to generate the waves for *p* and *s* polarization, reaching the air/prism interface at external angle of incidence α. The light beam is then coupled into the SPR structure using the equilateral prism with the angle of incidence θ given by relation $\theta \left(\lambda \right)=60^{\circ }-\sin^{-1}\left[\sin\alpha /n_{BK7}\left(\lambda \right)\right]$, where $n_{BK7}\left(\lambda \right)$ is the wavelength-dependent RI of the prism. The reflected light then goes through an analyzer oriented $0^{\circ }$ or $90^{\circ }$ with respect to the plane of incidence to generate the reflectances for either *p* or *s* polarization.

The process of measurement consists of multiple steps. Firstly, using the optical fiber, the light is guided through the collimating lens so that the collimated light beam goes through the polarizer oriented $45^{\circ }$ with respect to the plane of incidence. Generated *p* and *s* components are then coupled by the equilateral prism into the SPR structure, which is attached to the rotary desk, allowing us to adjust different angles of incidence. For the *p*-polarized component, the ATR takes place at the prism/structure interface and the reflected light beam then goes through the analyzer. In the first part of the measurement, the analyzer is set to be $90^{\circ }$ with respect to the plane of incidence to generate the spectrum $I_{s}^{ref}\left(\lambda \right)$. Initially, the spectrum is captured for air when no analyte is present. In the second part of the measurement, the analyte is applied, and the spectrum $I_{p}\left(\lambda \right)$ is captured for the analyzer oriented $0^{\circ }$ with respect to the plane of incidence. The corresponding obtained reflectance ratio is given as $R_{p}\left(\lambda \right)/R_{s}^{ref}\left(\lambda \right)$. The measurements of spectral reflectance ratio $R_{p}\left(\lambda \right)/R_{s}^{ref}\left(\lambda \right)$ for aqueous solutions of NaCl with concentrations in a range of 0–10 wt% with a step of 2 wt% were performed at a temperature of 22.8 ${}^{\circ }$C, which was kept constant during the experiment. The analyte RIs $n_{D}$ at a wavelength of 589 nm (sodium D line) were measured by a digital refractometer (AR200, Reichert). The RIs were 1.3331, 1.3358, 1.3427, 1.3492, 1.3587, and 1.3599.

### Experimental Results and Discussion

Measured spectral reflectance ratios $R_{p}\left(\lambda \right)/R_{s}^{ref}\left(\lambda \right)$ for the external angle of incidence $\alpha =-15.4^{\circ }$ ($\theta \approx 70.2^{\circ }$) for both structures are shown in Figure 9. It can be observed that with the increasing thickness of silicon dioxide, the resonance shifts towards longer wavelengths.

The resonance wavelength as a function of the refractive index of the analyte and the corresponding second-order polynomial fit for both structures is shown in Figure 10a. It is evident that the shift in the resonant wavelength is approaching a value of 200 nm for the different refractive indices of the aqueous solutions of NaCl.

Specifically, for the refractive index values of the analyte ranging from 1.3331 to 1.3599, the resonant wavelengths in the range of 1226.7–1406.1 nm were obtained for the structure with the silicon dioxide layer thickness $t_{{\mathrm{SiO}}_{2}}=147.5$ nm, and the resonant wavelengths in the range of 1539.4–1731.3 nm were obtained for the structure with the silicon dioxide layer thickness $t_{{\mathrm{SiO}}_{2}}=270.9$ nm. The achieved RI sensitivity at the angle of incidence $\theta \approx 70.2^{\circ }$ for both structures is shown in Figure 10b. For the first silicon dioxide overlayer thickness, the achieved RI sensitivity varies in the range of 4505–8140 nm/RIU, and for the second silicon dioxide overlayer thickness in the range of 5305–8690 nm/RIU.

Measured spectral reflectance ratio $R_{p}\left(\lambda \right)/R_{s}^{ref}\left(\lambda \right)$ for a smaller angle of incidence was extended for the silicon dioxide layer thickness $t_{{\mathrm{SiO}}_{2}}=147.5$ nm only, because, for the silicon dioxide layer thickness $t_{{\mathrm{SiO}}_{2}}=270.9$ nm, some of the dips were outside of the measurement range of the spectrometer, and the results are shown in Figure 11a. For the refractive index values of the analyte ranging from 1.3331 to 1.3599, the resonant wavelengths in the range of 1486.3–1816.9 nm were obtained. The shift in resonant wavelength for the different refractive indices of the aqueous solutions of NaCl attains a value of 330.6 nm. The resonance wavelength as a function of the refractive index of the analyte and the corresponding second-order polynomial fit are shown in Figure 11b, and the fit agrees well with the theoretical dependence also shown in the same figure. The achieved RI sensitivity is shown in comparison to the theoretical values calculated for the angle of incidence $\theta =65.6^{\circ }$ in Figure 12a. The achieved RI sensitivity varies in the range of 8570–15,785 nm/RIU, and the highest figure of merit attains a value of 37.9 RIU${}^{-1}$.

The normalized optical field intensity distribution in the analyte (water) at resonance wavelengths for different silicon dioxide layer thicknesses $t_{{\mathrm{SiO}}_{2}}$ and angles of incidence θ is shown in Figure 12b. It is evident that the penetration depth [37] of the optical field in the analyte (evanescent tail) is larger for the thicker silicon dioxide overlayer, thus justifying the higher sensitivity. Moreover, an increase in both the penetration depth in the analyte and sensitivity can be achieved by adjusting the angle of incidence, as demonstrated in Figure 12a,b for the silicon dioxide layer thickness $t_{{\mathrm{SiO}}_{2}}=147.5$ nm and the angle of incidence $\theta =65.6^{\circ }$.

## 7. Conclusions

In this paper, highly sensitive plasmonic structures of a simple design were employed in the Kretschmann configuration for liquid analyte sensing. Two structures, comprising a glass substrate with a gold layer, two adhesion layers of chromium, and a silicon dioxide overlayer with distinct protective layer thicknesses, were analyzed theoretically and experimentally. The thicknesses of individual layers were determined by the method of spectral ellipsometry. The sensing properties of both structures were derived based on the wavelength interrogation method, utilizing the spectral reflectance response of the SPR structures.

Theoretical spectral reflectances for aqueous solutions of NaCl with distinct concentrations were calculated for both structures, and the derived RI sensitivities varied within the ranges of 5150–7130 nm/RIU and 5720–7660 nm/RIU for the first and second structures, respectively. The sensitivity was also expressed in terms of the mass fraction of the solute and they varied within the ranges of 7.9–10.8 nm/wt% and 8.7–11.6 nm/wt% for the first and second structures, respectively. The highest FOM calculated attained a value of 38.0 RIU${}^{-1}$. Furthermore, measurements of the spectral reflectance ratio for aqueous solutions of NaCl with distinct concentrations were performed for both structures. Achieved RI sensitivities varied within the ranges of 4505–8140 nm/RIU and 5305–8690 nm/RIU for the first and second structures, respectively. Extending the measurements for a smaller angle of incidence, the RI sensitivity of 8570 to 15,785 nm/RIU, and the FOM of 37.9 RIU${}^{-1}$, were reached.

The main advantage of the structure lies in its simple design. The protective layer of silicon dioxide ensures the repeatability of measurement and the durability of the structure. It enables also polymer diffractive structures to be included [38]. The use of the structure is applicable to a wide variety of both gaseous and aqueous analytes, where even more aggressive environments can be considered. Furthermore, based on the measurements, the structures exhibit high sensitivity to the refractive index of the analyte, which can be adjusted by choosing a suitable angle of incidence.

## Figures and Tables

**Figure 1 nanomaterials-12-03090-f001:**
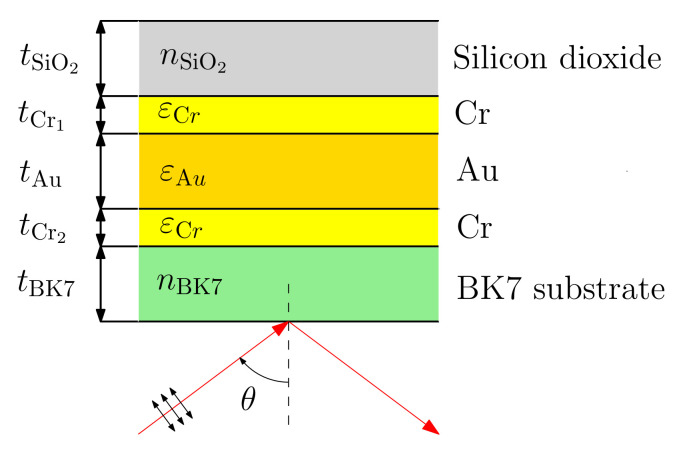
Schematic drawing of a multilayer structure.

**Figure 2 nanomaterials-12-03090-f002:**
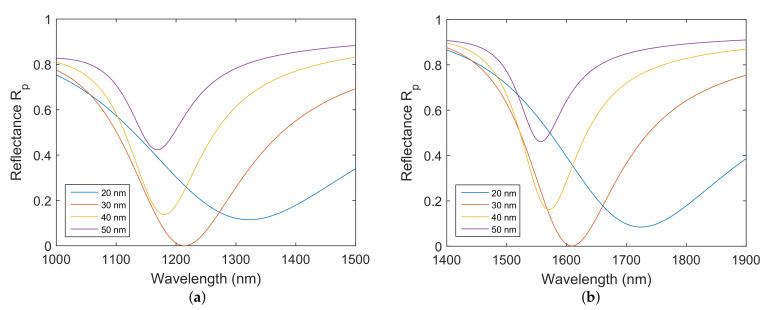
Theoretical spectral reflectance $R_{p}\left(\lambda \right)$ for distinct gold layer thicknesses for: $t_{{\mathrm{SiO}}_{2}}=150$ nm (**a**), $t_{{\mathrm{SiO}}_{2}}=300$ nm (**b**). Angle of incidence $\theta =68^{\circ }$. Chromium layer thicknesses $t_{\mathrm{Cr}}=2$ nm. The analyte is water.

**Figure 3 nanomaterials-12-03090-f003:**
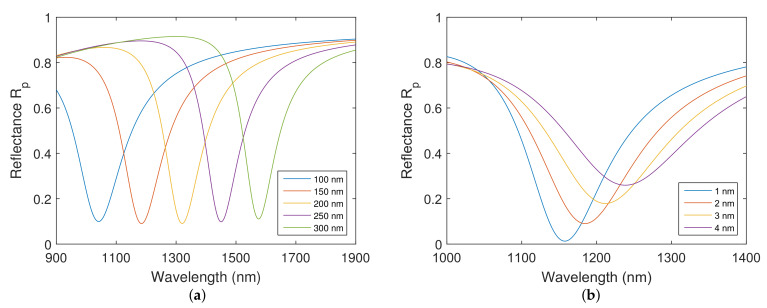
Theoretical spectral reflectance ratio $R_{p}\left(\lambda \right)$ for distinct silicon dioxide layer thicknesses (**a**) and for distinct adhesion chromium layer thicknesses (**b**). Gold layer thickness $t_{\mathrm{Au}}=38$ nm. Angle of incidence $\theta =68^{\circ }$. The analyte is water.

**Figure 4 nanomaterials-12-03090-f004:**
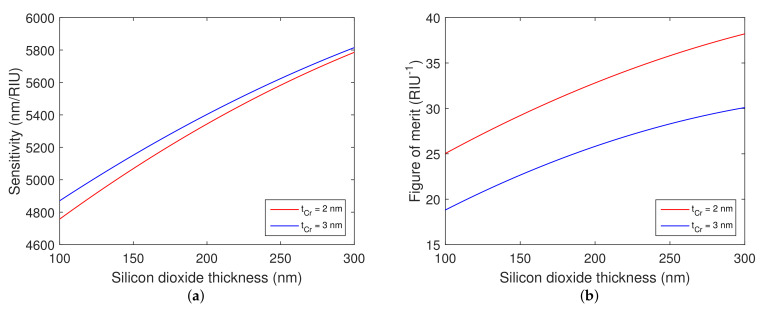
Theoretical RI sensitivity (**a**) and figure of merit (**b**) as a function of the silicon dioxide thickness when water was considered as an analyte. Gold layer thickness $t_{\mathrm{Au}}=38$ nm. Silicon dioxide thickness $t_{{\mathrm{SiO}}_{2}}=150$ nm. Angle of incidence $\theta =68^{\circ }$.

**Figure 5 nanomaterials-12-03090-f005:**
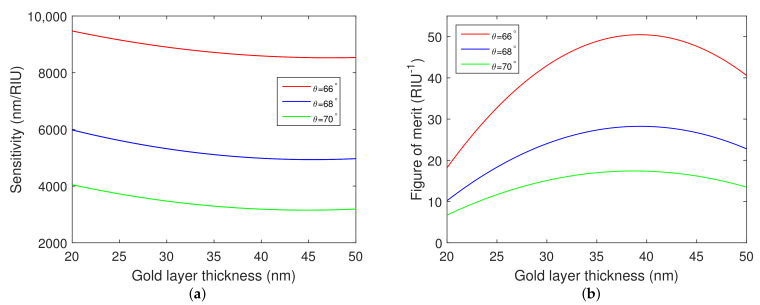
Theoretical RI sensitivity (**a**) and figure of merit (**b**) as a function of the gold layer thickness when water was considered as an analyte. Silicon dioxide thickness $t_{{\mathrm{SiO}}_{2}}=150$ nm. Chromium layer thickness $t_{\mathrm{Cr}}=2$ nm. Angles of incidence $\theta =66^{\circ }$, $68^{\circ }$, and $70^{\circ }$.

**Figure 6 nanomaterials-12-03090-f006:**
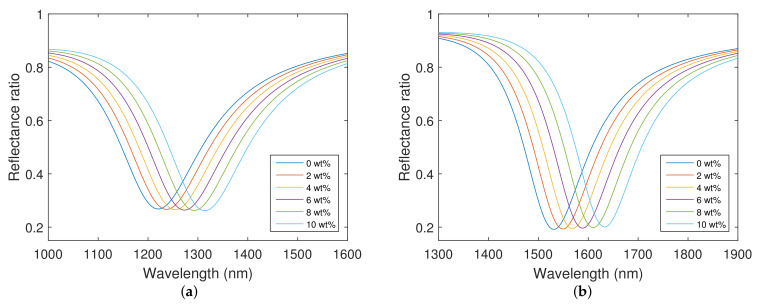
Theoretical spectral reflectance ratio $R_{p}\left(\lambda \right)/R_{s}\left(\lambda \right)$ for the silicon dioxide layer thicknesses: $t_{{\mathrm{SiO}}_{2}}=147.5$ nm (**a**), $t_{{\mathrm{SiO}}_{2}}=270.9$ nm (**b**). Angle of incidence $\theta =68^{\circ }$.

**Figure 7 nanomaterials-12-03090-f007:**
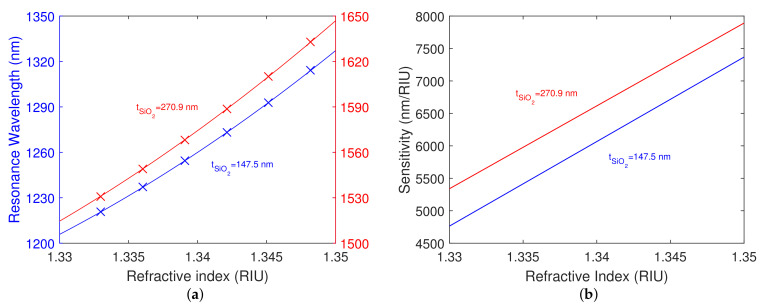
Theoretical resonance wavelength as a function of refractive index of the analyte and second-order polynomial fit for both structures (**a**). Theoretical RI sensitivity for both structures (**b**). Angle of incidence $\theta =68^{\circ }$.

**Figure 8 nanomaterials-12-03090-f008:**
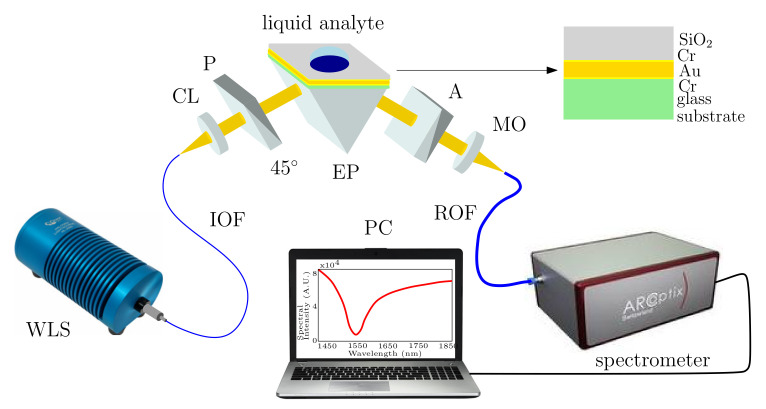
Schematic drawing of the experimental setup employing the SPR structure: white light source (WLS), input optical fiber (IOF), collimating lens (CL), polarizer (P), equilateral prism (EP), analyzer (A), microscope objective (MO), read optical fiber (ROF), and personal computer (PC).

**Figure 9 nanomaterials-12-03090-f009:**
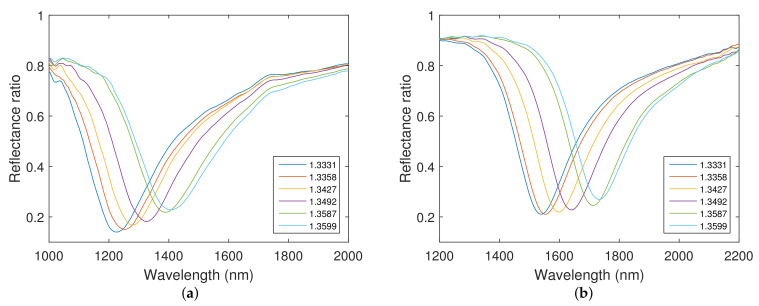
Measured spectral reflectance ratios $R_{p}\left(\lambda \right)/R_{s}^{ref}\left(\lambda \right)$ for the silicon dioxide layer thicknesses: $t_{{\mathrm{SiO}}_{2}}=147.5$ nm (**a**), $t_{{\mathrm{SiO}}_{2}}=270.9$ nm (**b**). The angle of incidence $\theta \approx 70.2^{\circ }$.

**Figure 10 nanomaterials-12-03090-f010:**
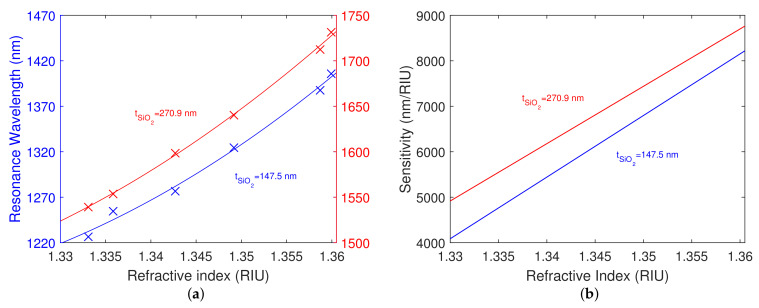
Measured resonance wavelength as a function of refractive index of the analyte and second-order polynomial fit for both structures (**a**). RI sensitivity measured for both structures (**b**). The angle of incidence $\theta \approx 70.2^{\circ }$.

**Figure 11 nanomaterials-12-03090-f011:**
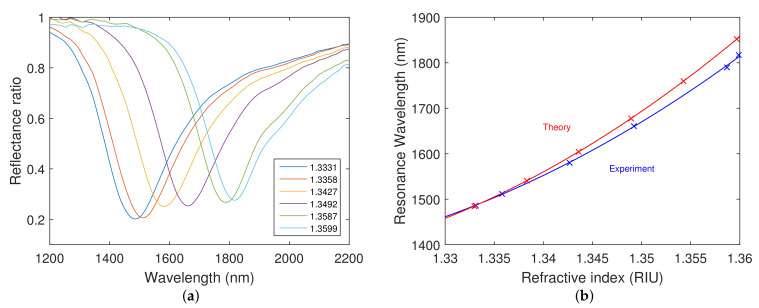
Measured spectral reflectance ratios $R_{p}\left(\lambda \right)/R_{s}^{ref}\left(\lambda \right)$ (**a**). RI sensitivity measured for a smaller angle of incidence (blue) and theoretical RI sensitivity for angle of incidence $\theta =65.6^{\circ }$ (red) (**b**). The silicon dioxide layer thickness $t_{{\mathrm{SiO}}_{2}}=147.5$ nm.

**Figure 12 nanomaterials-12-03090-f012:**
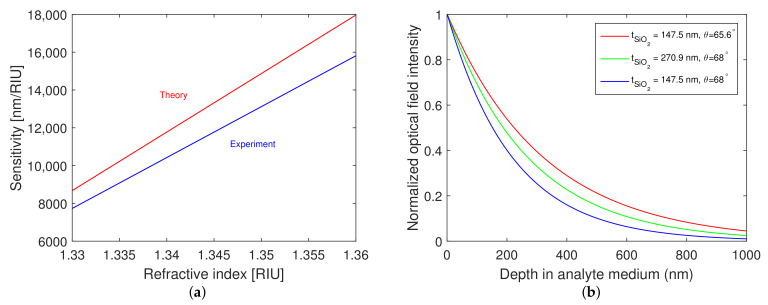
Comparison of RI sensitivity measured for a smaller angle of incidence (blue) and theoretical RI sensitivity (red) (**a**). Normalized optical field intensity distribution in the analyte (water) for different silicon dioxide layer thicknesses $t_{{\mathrm{SiO}}_{2}}$ and angles of incidence θ (**b**).

**Table 1 nanomaterials-12-03090-t001:** Parameters for the Drude–Lorentz dispersion formula.

Drude Term Parameter	Value	Oscillator 1 Parameter	Value	Oscillator 2 Parameter	Value
$\epsilon_{\infty }$	1	$A_{1}$	3.612	$A_{2}$	1.423
$\lambda_{p}$ (nm)	133.85	$\lambda_{1}$ (nm)	309.11	$\lambda_{2}$ (nm)	424.06
$\gamma_{p}$ (nm)	27851	$\gamma_{1}$ (nm)	2591.3	$\gamma_{2}$ (nm)	1515.2

**Table 2 nanomaterials-12-03090-t002:** Parameters for the Drude CP dispersion formula.

Drude Term Parameter	Value	CP Term 1 Parameter	Value	CP Term 2 Parameter	Value
$\epsilon_{\infty }$	1.129	$A_{1}$	33.086	$A_{2}$	1.659
$\lambda_{p}$ (nm)	213.67	$\lambda_{1}$ (nm)	1082.3	$\lambda_{2}$ (nm)	496.5
$\gamma_{p}$ (nm)	4849.8	$\gamma_{1}$ (nm)	1153.2	$\gamma_{2}$ (nm)	2559.7
		$\Phi_{1}$ (nm)	−0.25722	$\Phi_{2}$ (nm)	0.83533

**Table 3 nanomaterials-12-03090-t003:** The thickness parameters of the individual layers for both structures.

$t_{{\mathrm{SiO}}_{2}}$	$t_{{\mathrm{Cr}}_{1}}$	$t_{\mathrm{Au}}$	$t_{{\mathrm{Cr}}_{2}}$	$t_{\mathrm{BK}7}$
147.5 nm	3.66 nm	38.95 nm	2 nm	1 mm
270.9 nm	3.11 nm	37.93 nm	2 nm	1 mm

## Data Availability

Not applicable.

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
