# Peer review of "Highly Sensitive Plasmonic Structures Utilizing a Silicon Dioxide Overlayer"

_nanomaterials, 2022, doi:10.3390/nano12183090_

Round 1

Reviewer 1 Report

Review: In this paper, the authors present a simple ultrahigh-sensitive plasmonic structures are analyzed theoretically and experimentally. The structure comprising a glass substrate with a gold layer, two adhesion layers of chromium, and a silicon dioxide overlayer, is employed in liquid analyte sensing. 

1. The authors claim that the proposed sensor is ultra-sensitive. However, there is no comparison between the performance of the proposed sensor and its counterparts. I recommend the authors compare the proposed structure with the state of the art (please see below).

·         Cetin, A. E. et al. Plasmonic nanohole arrays on a robust hybrid substrate for highly sensitive label-free biosensing. ACS Photon. 8, 1167–1174 (2015).

·         Sreekanth, K. V. et al. Extreme sensitivity biosensing platform based on hyperbolic metamaterials. Nat. Mater. 15, 621–627 (2016).

2. The whole design procedure of the structure sounds a bit ambiguous, for instance, how the structure is designed and how the thicknesses and sizes are chosen are not discussed, and the manuscript offers no justification for choosing these geometrical parameters.

I recommend the authors comment on how the thickness of the SiO2 is defined in the current structure.

3. Are there any limitations to increasing the thickness of SiO2?  For example, do you think increasing SiO2 thickness will saturate? To achieve reasonable results, what is the minimum thickness one can choose? How varying the thickness of the Cr layer affects the performance of the proposed sensor? (Variations in thickness of the Cr layer clearly affect the near-field and also the FWHM of the reflection spectra. Since the proposed sensor is based on SPP, it should be sensitive to changes in the near-field and evanescent tails of the SPPs).

4. The thickness of Cr/Au/Cr is different in both structures. I understand this can be a fabrication imperfection, but the authors must address it.  Comparing the simulation results for the tSiO2=147.5nm and using the Cr/Au/Cr thickness for the second structure (and vice versa) might build a stronger case for avoiding the fabrication imperfection.

Recommendation to the authors

1. Without comparing the paper to the state of the art, the proposed title should be changed

2. I recommend the authors add a concept schematic in Fig. 1 with the incident beam, its propagation direction, and polarization.

3. In Fig. 3 and Fig. 4, I recommend the authors use different colors for each angle and add a legend.

4. In Fig. 7, it would be better to draw both figures on the same vertical scale so that one can easily compare the two cases.

Reviewer 2 Report

In the presented manuscript, a fragmented theoretical and experimental study of the characteristics of surface plasmon structure with top dielectric layer is described.

1. Line 60. Thickness of the first structure is given, but that of the second structure is not discussed.

2. As ellipsometric data evaluation is discussed in Section 2.5, the comments on the ways of obtaining of the real and imaginary parts of ro (Eq. 1) from the measurements of the reflected intensities are desirable.

3. Lines 189-192. There is no any evidence or prove presented that larger field penetration depth results in the higher sensitivity. If authors analyze their data on field distribution presented on Figs. 7 and 12(b) and sensitivity on Fig. 6 and 12(a), they can discover the opposite behavior.

4. It is not clear why authors concentrated on two particular thicknesses of SiO2. It would be beneficial to present theoretical analysis of dependence of FOM on SiO2 and metal thickness.

5. From the description of optical setup shown in Fig. 8, the necessity and role of the polarizer P is not clear.

6. It is not clear why authors used the ratio of the reflectance values Rp/Rs, were Rp and Rs normalized to intensities of the incident beam of the corresponding polarizations. In general, the ratio of the reflectivities of the reflected and incident beams is sufficient for optical characterization.

7. Line 252. It is not clear, why authors took this precise wavelength value? Units are missed.

8. Lines 281-283. Sensitivity itself does not characterize the lower detection limit. The broad resonances result in a low resolution that can be characterized by FOM, which is quite low as compared to waveguide sensors for example. The Introduction does not discuss the resolution of sensing structures that support other types of normal optical modes. Discussion of sensitivity of the resonance wavelength only does not provide any idea on sensing performance. The resolution of the presented structure is low, and it cannot be referred to as ultrahigh-sensitive in the text.

Reviewer 3 Report

This paper reports theoretical and experimental results on a simple desing for a SPR sensor based on a multilayer where a top layer of SiO2 is used to increase the sensitivity.

The idea of using a top layer on SPR sensor is not new and several reports can be find in literature. here it seems that the sensitivity reached a very high value.

Anyway, the manuscript presents several criticisms that need to be solved / improved before to be considered for publication:

1. methods section is full of basic and not-useful information. in my opinion is not interesting at all to report model for the elipsometry (very well known) or the models for data fitting (section 2.2 and section 2.4 and 2.5)

2. The resonance used for the SPR is in the NIR region where water is not transparent. Clearly the analyses are done in reflection, but I expect some effect on the measured spectra

3. fig 7, the field intensity of 11 and 5 are at the gold interface ? from the figure is not clear. To me it seems that the field is at metal / SiO2 interface, while at SiO2 / water the field is much less. These simulations are quite unuseful considering that they cannot explain the high sensitivity (the field enhancement is very low)

4. The most critical issue is related to the presented sensitivity. from the results reported in figures 6, 9, 10, 11 and 12 it seems that the sensitivity increases increasing the refractive index of the analyte! this is absolutely weird. The sensitivity of a biosensor is defined (in the case of a SPR sensor) as the change in resonance position (nm) over the change in refractive index (RIU) but it MUST BE constant to be correct on the contrary the definition is not valid anymore. 

The authors must check this and correct reporting a correct estimation of the sensitivity. 15.000 nm /RIU are close to the top sensitivity reported with optimized plasmonic devices and I am rather sure that it cannot be possible to achieve it with a SPR prism sensor.

I recommend also to mention the top performers as experimental plasmonic SPR sensors :

Nat. Mater., 2009, 8, 867.
Nanoscale Horiz., 2019,4, 1153-1157
Nat. Mater., 2016, 15, 621–627.
Photonics Research Vol. 10, Issue 1, pp. 84-95 (2022)

Round 2

Reviewer 1 Report

The authors responded to all my questions.

Author Response

No comments.

Reviewer 2 Report

In the first revision, the authors have addressed a number of questions. However, there are few important points remained to be revised.

Previous comments

1.3. “… There is no any evidence or prove presented that larger field penetration depth results in the higher sensitivity. …”

As no theoretical prove or a reference to a theoretical study were not given, I suggest to remove this context.

1.4. "... It would be beneficial to present theoretical analysis of dependence of FOM on SiO2 and metal thickness."

It was not a request for calculations of the reflectivity spectra. It is the request for analysis of FOM dependences on the structural parameters.

New comment.

2.1. The external angle of incidence alpha was not introduced and demonstrated. It is not clear, why authors need the discussion of both the internal and external angles of incidence. If the dispersion of prism refractive index is assumed in the experiments, it is very easy to recalculate the internal angle of incidence in theoretical studies by the 2x2 matrix method using Eq. (5). Moreover, in experiments, the BK7 dispersion changes the sensing characteristics that should be estimated. In this case, all the data should be presented for the external angle of incidence.

Reviewer 3 Report

I think that the manuscript is partially improved, but there is still a major issue that I really cannot understand

the authors say that the sensitivity can be a function of the refractive index of the analyte. This is ABSOLUTELY WRONG. the definition of sensitivity, as reported in hundreds of papers, is the shift in wavelength with respect to the index of refraction and MUST be a linear function! consequently, the sensitivity (nm/RIU) must be constant!

the authors must correct this or better explain their results supporting them with some literature

Round 3

Reviewer 2 Report

Comments on the reply to Comment 1.4.

1.4. "... It would be benecial to present theoretical analysis of dependence of FOM on SiO2 and metal thickness.”

It was not a request for calculations of the reflectivity spectra. It is the request for analysis of FOM dependences on the structural parameters.

The reflectivity spectra are important from the point of view of the effect of the structural parameters. Because the optimal metal thickness was obtained, it is desirable to analyse FOM dependences on the silicon dioxide thickness, including two different thicknesses of the adhesion chromium layer.

According to the title of the manuscript optimization of the structure is performed based on the condition of achievement of highest sensitivity. Although, authors introduce FOM that characterizes the possible resolution, the text does not describe the procedure based on FOM maximization. In this case, it is not clear, why authors need FOM and sensitivity parameters introduced in Eqs. (9) and (8), if their choice of structural parameters is based on uncpecified phenomenological assumptions. Summarizing, the manuscript does not provide even a simple study of sensitivity behavior on the structural parameters like, for example, it was done in similar study of MIM structures [Ann. Phys. (Berlin) 2018, 1700411].

Instead of presenting similar Figures 2a and 2b, which do not give any idea on the maximal resolution of the sensing structure, FOM dependences on the metal layer and SiO2 thicknesses, as well as on the incidence angles will be sufficient for understanding the behavior of resolution.

Author Response

The authors would like to thank Reviewer 2 for his suggestions and comments to help improve our work.

Following the recommendation, RI sensitivity and FOM as a function of the gold layer thickness were added in Figs. 5(a) and (b). The definition of FOM was expanded so that the depth of dip D is also considered. Consequently, the FOM values were changed to satisfy the new relation.

Reviewer 3 Report

I think that the manuscript is now ok for publication

Author Response

No comments.

Round 4

Reviewer 2 Report

The manuscript can be accepted for publication in the present form.